# An Optimization-Based Framework to Dynamically Schedule Hospital Beds in a Pandemic

**DOI:** 10.3390/healthcare13182338

**Published:** 2025-09-17

**Authors:** Marwan Shams Eddin, Hussein El Hajj

**Affiliations:** 1Department of Systems Engineering and Operations Research, George Mason University, Fairfax, VA 22030, USA; 2Department of Information Systems and Analytics, Santa Clara University, Santa Clara, CA 95053, USA; helhajj@scu.edu

**Keywords:** pandemic, hospital bed scheduling, resource sharing, robust optimization, healthcare operations

## Abstract

**Background:** Emerging pandemics can rapidly overwhelm hospital capacity, leading to increased mortality and healthcare costs. **Objective:** We develop an optimization-based framework that dynamically schedules hospital beds across multiple facilities to minimize total healthcare costs, including patient rejections and logistical expenses, under resource constraints. **Methods:** The model integrates several real-world flexibilities: standard hospital beds, buffer capacity from non-pandemic wards, in situ field hospitals, and inter-hospital patient transfers. To capture demand uncertainty, we link the model with an SEIRD epidemic forecasting approach and further extend it with a robust optimization variant that safeguards against worst-case surges. Recognizing computational challenges, we reformulate the problem to significantly reduce solution times and derive structural properties that provide guidance on when to open field hospitals, allocate buffer beds, and prioritize patients across facilities. **Results:** A case study based on COVID-19 data from Northern Virginia shows that the proposed framework reduces healthcare costs by more than 50% compared with current practice, mainly by lowering patient rejection rates. **Conclusions:** These results highlight the value of combining epidemic forecasting with prescriptive optimization to improve resilience and inform healthcare policy during crises.

## 1. Introduction and Motivation

The COVID-19 pandemic exposed the fragility of hospital capacity under prolonged public health emergencies [1]. In many countries, surging infections quickly overwhelmed intensive care units (ICUs) and medical wards, forcing healthcare systems to operate in crisis mode [2,3,4]. At the peak on 12 April 2020, for example, over 18,800 hospital beds in England were occupied by COVID-19 patients [5]. Similar scenarios played out globally, with hospitals struggling to provide care for both COVID-19 and non-COVID patients [4,5]. When demand exceeds supply, healthcare providers face important decisions: how to allocate scarce beds, ventilators, and staff to save the most lives. Some regions resorted to formal triage policies and ethical guidelines to prioritize patients for critical care, aiming to maximize survival and social well-being under resource constraints [6,7]. In practical terms, this meant that access to care could not be guaranteed for all; many hospitals were forced to reject patients or delay treatment for those with lower survival probability, a costly outcome in both human and economic terms [8]. The need to minimize patient rejections while managing limited resources has become a paramount operational challenge in pandemic response.

To avert worst-case outcomes, health systems worldwide implemented extraordinary measures to expand capacity and streamline patient flows [9,10]. Elective surgeries were canceled altogether to free up beds, and dozens of field hospitals (“in situ” hospitals) were erected in convention centers, sports arenas, and other large venues to absorb overflow demand [5]. For instance, the UK’s National Health Service rapidly established Nightingale hospitals, and U.S. cities converted conference centers into COVID-19 wards. These steps, combined with procuring emergency equipment and redeploying staff, were critical to enhance surge capacity [5]. At the same time, policymakers recognized that better coordination across healthcare facilities could greatly improve system resilience [1]. Rather than letting each hospital operate in isolation, some regions set up centralized coordination centers to balance loads and facilitate inter-hospital transfers [11,12,13]. For example, the state of Minnesota established a Medical Operations Coordination Center to redistribute ICU patients across hospitals with available beds [14]. This kind of collaboration, effectively sharing resources and patients, can ensure that a critically ill patient turned away from one overfilled hospital can be transferred to another facility with capacity. Indeed, such load-balancing mechanisms proved essential for preserving critical care access during peak surges [14,15]. Without them, many beds in less affected areas might remain idle while patients in hot spots are denied care.

In this paper, we develop an integrated optimization framework for dynamic hospital bed allocation in a pandemic. At its core, our approach is a multi-period, multi-hospital resource allocation model that minimizes total healthcare costs under capacity constraints. A central feature is that the objective function attaches a heavy penalty to unmet demand, where essentially, there is a cost of patient rejection, reflecting the severe health and societal consequences of turning away sick patients. By internalizing this penalty, the model actively strives to avoid scenarios in which patients in need are denied access to beds. Meanwhile, the model also accounts for operational costs of care (for regular and ICU beds in both standard hospitals and field hospitals) as well as logistical costs for moving patients. The proposed framework empowers a coordinated, system-level strategy: a healthcare agency makes strategic decisions on how many patients to allocate to each hospital or pop-up facility, and then hospitals make operational decisions on how to redistribute those patients via transfers or supplementary beds.

We incorporate multiple real-world flexibilities, which include the use of buffer beds (temporarily repurposed non-pandemic beds), the opening of in situ field hospitals to boost capacity, and the option to transfer patients between hospitals and field units. By optimally timing and sizing these interventions, the model can expand effective capacity where it’s most needed, alleviating bottlenecks at overwhelmed sites. Notably, we formulate the patient transfer network with a novel flow-based representation that aggregates inter-hospital transfers through a dummy exchange node. This reformulation is mathematically equivalent to modeling every pairwise transfer, but it dramatically reduces computational complexity. In fact, it cuts the number of decision variables and constraints by an order of magnitude, yielding over a 90% reduction in solution time for large-scale instances. Such tractability is important for practical use, as a regional health authority might be coordinating dozens of hospitals over many weeks, which represents a problem size intractable for a naive formulation.

Another distinctive aspect of our approach is the integration of epidemic dynamics and robust optimization into the resource planning process. We link our allocation model with a compartmental SEIRD (Susceptible–Exposed–Infected–Recovered–Deceased) model to forecast the trajectory of infections and hospitalizations over the planning horizon. This provides a data-driven estimate of the time-varying demand for hospital beds, stratified by patient severity (e.g., mild vs. severe cases). However, rather than taking these forecasts at face value, we acknowledge the considerable uncertainty in pandemic demand. We therefore formulate a robust optimization variant of the model that hedges against worse-than-expected infection scenarios. In the robust model, the demand in each period is allowed to vary within an uncertainty set (e.g., a confidence interval around the SEIRD prediction), and we seek a solution that minimizes the worst-case total cost. This yields a hedging strategy that may deliberately hold extra surge capacity or preemptively distribute patients more conservatively in order to cushion the impact of a potential demand spike. Importantly, we prove that this robust planning problem can be solved just as efficiently as the deterministic case, essentially by solving a variant of the model with demand set to its upper-bound scenario. In summary, our framework combines predictive modeling (epidemic forecasts) with prescriptive analytics (optimization under uncertainty), enabling policymakers to anticipate demand surges and allocate resources proactively rather than reactively.

We demonstrate the effectiveness of the proposed approach through both theoretical insights and a detailed case study. First, we derive analytical structural properties of the optimal policy in certain regimes. For instance, when patient demand persistently exceeds total capacity, we show that an optimal strategy will utilize all available resources to the fullest, where all hospitals and field units will operate at maximum capacity (including using every buffer bed), as any idle capacity would incur unnecessary shortage costs. Conversely, if overall demand never exceeds supply, the model naturally prioritizes the lowest-cost resources: hospitals are ranked by their per-patient operating cost, and the optimal solution is to fill cheaper hospital beds first (while still ensuring all demand is met), activating higher-cost facilities like field hospitals only if needed. These intuitive yet non-trivial properties provide reassurance about the model’s behavior and offer guidance for decision-makers (e.g., when to trigger the opening of an in situ hospital or how to distribute buffer beds across hospitals in a worst-case surge). We next apply our model to a real-world scenario based on Northern Virginia (NOVA) during the late-2021 COVID surge. This region experienced a severe capacity crunch; in particular, hospitalizations jumped by over 1000% within two months [16], pushing emergency rooms to their limits, and authorities identified several sites (conference centers, a university arena, etc.) for potential field hospitals. Using data on NOVA’s hospital network, COVID infection rates, and available surge facilities, we compare the performance of our optimized scheduling policy against the current practice. The results show striking improvements: Our model reduces total healthcare costs by more than 50% compared with current practice, primarily by sharply cutting the number of patients who must be turned away.

In the optimized solution, virtually no COVID-19 patients with severe symptoms are denied a hospital bed, whereas under the decentralized reactive approach, hundreds of such patients would go untreated at the peak of the wave. This outcome is achieved by a timely reconfiguration of capacity, where the model advises, for example, opening a large field hospital in anticipation of the surge and judiciously using available buffer beds in urban hospitals, as well as a more selective admission policy for mild cases when critical care space is running low. Interestingly, we find that explicit coordination via patient transfers plays a complementary role: by reallocating patients from overcrowded hospitals to those with spare capacity, the system avoids unnecessary bottlenecks and improves overall utilization. Our case study also evaluates the robust optimization strategy under stochastic demand scenarios. Even if the future unfolds worse than the average forecast, the robust plan maintains superior performance, reducing worst-case costs by 39% relative to a non-robust (average-based) plan and markedly limiting the variability of outcomes. In practical terms, this means greater assurance that the health system can withstand surprise surges without resorting to crisis standards of care. These findings underscore the value of proactive, model-driven planning in pandemic management. That is, by integrating epidemiological insight, system-wide coordination, and flexible capacity deployment, our models show significant gains in efficiency, equity, and resilience.

The contributions of this work are therefore both methodological and practical. We introduce a novel optimization model for pandemic bed management that captures the key complexities of the problem, including multi-tier decision-making, resource sharing, and uncertainty, in a tractable formulation. We provide an equivalent reformulation and analysis that make it feasible to solve large regional instances and to derive clear policy rules. Finally, we translate the model’s prescriptions into real-world impact through a case study, offering evidence-based guidance for health planners. While motivated by COVID-19, our framework is generalizable to future health crises, whether another infectious disease outbreak or any scenario causing a sudden mismatch between patient demand and hospital supply. The proposed approach minimizes the human and economic costs of such mismatches, which can help health systems ensure access to care even in the face of unprecedented surges, avoiding the tragic consequences of uncoordinated or insufficient responses.

In light of these challenges, it is important to situate our work within the broader body of research on hospital bed scheduling and pandemic resource management. The challenge of scheduling hospital beds and optimizing resource management during periods of high demand, such as pandemics, has been extensively explored in the literature. To provide a structured overview, we categorize the relevant research into two main streams. The first focuses on the optimal allocation of hospital beds and ICUs for both pandemic and non-pandemic patients. The second, on the other hand, examines broader resource management strategies and resource sharing opportunities during pandemics to improve the overall efficiency of the healthcare system.

Allocating pandemic and non-pandemic beds and accepting patients is critical to reducing healthcare costs. Several predictive approaches have been developed to estimate hospital demand during pandemics, ranging from time-series forecasting and machine learning to simulation models [2,3,4,7,17,18]. These methods provide valuable projections of future case loads, but they are primarily descriptive and must be complemented by prescriptive optimization frameworks to support actionable resource allocation decisions. Optimization-based studies have designed admission and scheduling controls using methods such as Markov decision processes, dynamic programming, and integer programming [19,20,21]. Other approaches include invitation policies for proactive service systems [22] and disposition strategies to generate surge capacity in hospitals [23]. Other studies extended this stream with a dynamic programming model that admits multiple patient classes (COVID-19, emergency, and elective care) [24]. Multi-hospital settings have also been examined [25,26], though often without coordination. A key limitation of these studies is the assumption that healthcare entities and hospitals operate independently, overlooking significant opportunities for flexibility and capacity expansion.

Another relevant stream concerns resource utilization and sharing in pandemics. Studies have examined the allocation of vaccines, treatments, testing kits, ventilators, and hospital beds. For instance, vaccine distribution under supply constraints has been modeled using SIR-type epidemic dynamics [27,28,29], while treatment allocation has been addressed via logistics-based nonlinear integer programs [10,30]. Screening and testing-kit allocation strategies have also been developed [31]. Beyond these specific resources, broader frameworks consider multiple resource types, including facilities, staffing, and transportation [9]. Critical medical equipment, such as ventilators, has received special attention, with optimization models informed by predictive demand forecasts [32,33]. For a review of these topics, see [1]. Closer to our setting are studies integrating epidemic demand forecasting with hospital bed allocation [10,15,34]. However, these works typically assume that strategic-level admission and allocation decisions cannot be redistributed, thereby overlooking opportunities for coordination and resource sharing (see, e.g., [11,12,13,35]). To the best of our knowledge, no prior work has jointly matched patient demand and bed supply during pandemics while simultaneously incorporating prioritization, patient transfers, use of non-pandemic beds, and in situ hospitals as capacity-expansion options. What further distinguishes our framework is its integration of epidemic forecasts into a robust optimization model, yielding immune solutions that minimize worst-case healthcare costs.

Table 1 summarizes representative studies from the literature, highlighting their settings, resources considered, methodologies, and limitations, and contrasting them with the distinct features of our proposed framework.

The remainder of the paper is organized as follows. In Section 2, we define the model, notation, optimization problem, proposed equivalent reformulation, structural properties, a robust optimization extension, and patient demand forecasting using an SEIR model. In Section 3, we demonstrate the effectiveness of our framework through a case study based on the COVID-19 outbreak during 2021–2022. Finally, in Section 4, we conclude and propose future directions.

## 2. Model Development

In this section, we develop an optimization-based framework for dynamically allocating hospital beds during an emerging pandemic, aiming to minimize total healthcare costs and alleviate strain on hospital capacity.

Consider an emerging pandemic in a population of size *N* in which the number of infections at time t∈T is denoted by $I_{t}\leq N$, where T={1,⋯,T} is the considered time horizon. Infected individuals exhibit varying levels of symptom severity, denoted by i∈I, where I represents the set of severity levels of symptoms ranging from mild to severe cases. As such, we write(1)$$I_{t}=\sum_{i\in I}I_{i,t},\;\forall t\in T,$$In many instances, both mild and severe cases may require hospitalization. For example, individuals with compromised health, such as the elderly or those with pre-existing respiratory conditions (e.g., asthma, bronchiectasis, or chronic obstructive pulmonary disease), can experience worsening symptoms even from mild infections, thereby increasing their risk of hospitalization [36]. To that end, we define the hospitalization demand of subjects having symptom severity level *i* at time *t* as $d_{i,t}$ defined as(2)$$d_{i,t}=I_{i,t},\forall t\in T,\;\forall i\in I,$$Surges in $d_{i,t}$ can cause shortages in hospital beds and hence can result in catastrophic healthcare costs. This highlights the urgent need for a well-structured scheduling plan to accommodate the resulting high demand. In what follows, we introduce a multi-period bed scheduling model designed to manage and accommodate patient surges effectively.

In this paper, we consider a two-stage decision-making process. The first is strategic and aims to allocate the patient demand $d_{i,t}$ over the system supply of hospital beds. Such decisions are often made by healthcare agencies that are in charge of tracking the number of infections and the bed utilization. These decision variables are denoted by $x_{i,t}^{p}$ and $x_{i,t}^{k}$, which define the initial number of patients allocated to a hospital p∈Ψ and to an in situ hospital k∈Ω. To that end, we denote by $s_{i,t}$ the shortage variable for patients of type i∈I at time *t*, which can be written as$$s_{i,t}=d_{i,t}-\sum_{p\in \Psi }x_{i,t}^{p}-\sum_{k\in \Omega }x_{i,t}^{k}\;\forall i\in I,\forall t\in T.$$We denote by $c_{i,t}$ the cost of a shortage, i.e., the cost of rejecting a patient who is in need of hospitalization. This cost is often high and might, in many cases, dominate all other logistical costs. Decisions must be adjusted accordingly to reduce shortages. In Section 2.2, we present theoretical results that characterize the optimal structure of the strategic bed allocation problem. We then describe the second level of decision-making, which addresses how to reassign patients across hospitals after the initial allocation in order to minimize logistical transfer costs.

We define the set Ψ as the set of hospitals, where for every hospital p∈Ψ, the associated bed capacity is denoted by $Q^{p}$. We track the bed utilization for a hospital *p* at time *t* using the state variable $H_{t}^{p}$ and associate it with $W_{t}^{p}$, which represents the cost of treating a patient. This cost may include expenses related to ICU units, ventilators, serum injections, staffing, operating costs, and other factors. In addition, in this paper, we consider that there exists a buffer of beds denoted as *B*. These beds are often borrowed from beds that are non-pandemic dedicated, i.e., beds for patients with other diseases. This means that *B* can be allocated to hospitals to boost their bed capacity. This can be modeled as follows:(3)$$\sum_{p\in \Psi }b_{t}^{p}\leq B,\;\forall t\in T\;\mathrm{and}\;H_{t}^{p}\leq Q^{p}+b_{t}^{p},\;\forall p\in \Psi ,\forall t\in T,$$
where $b_{t}^{p}$ is a decision variable that denotes the number of buffer beds allocated to hospital *p* at time *t*. Furthermore, in this paper, we allow for opening and operating in situ hospitals. As such, we define Ω as the set of locations at which in situ hospitals can be opened and operated. Similar to hospitals, an in situ hospital k∈Ω has a capacity $Q^{k}$ and a state variable that tracks the bed utilization and is denoted as $H_{t}^{k}$. The cost of treating and accommodating a patient in an in situ hospital is denoted as $W_{t}^{k}$. In addition to the costs for a hospital *p*, in this case, $W_{t}^{k}$ can also include infrastructure (setup, equipment, utilities, sanitation), operations (security, logistics), and administration (regulatory fees, IT) and is often higher than $W_{t}^{p}$ for a hospital *p*. The decision of opening and operating an in situ hospital is denoted as $u_{t}^{k}$, and this is often related to the availability of tents. As such, we have(4)$$\sum_{k\in \Omega }o^{k}u_{t}^{k}\leq O,\;\forall t\in T\;\mathrm{and}\;H_{t}^{k}\leq u_{t}^{k}Q^{k},\;\forall k\in \Omega ,\forall t\in T$$
where *O* is the available number of tents and $o^{k}$ is the number of tents that can be fit in an in situ hospital *k*. The constraint on the right guarantees that occupying an in situ hospital *k* can only occur when it is opened and operated, limited by its capacity $Q^{k}$. Each hospital p∈Ψ can transfer patients of type *i* at time *t* to another hospital q∈Ψ/{p} or to an opened in situ hospital k∈Ω, represented by $y_{i,t}^{p,q}$ and $z_{i,t}^{p,k}$, respectively. These transfers occur for various reasons, including differences in capacity, technology, staff availability, treatment capabilities, and operating costs. This flexibility in patient transfers helps optimize bed utilization and better accommodate patient demand. This directly means that the following holds:(5)$$\sum_{q\in \Psi }\sum_{i\in I}y_{i,t}^{p,q}+\sum_{k\in \Omega }\sum_{i\in I}z_{i,t}^{p,k}\leq H_{t}^{p},\;\forall p\in \Psi ,\forall t\in T.$$This basically guarantees that all patients transported from a hospital *p* cannot exceed the current number of patients $H_{t}^{p}$. The cost of transporting patients between hospitals *p* and *q* and between a hospital *p* and an in situ hospital *k* is indicated by $g^{p,q}$ and $g^{p,k}$. In general, these costs are relatively smaller than the other problem costs (e.g., shortage).

With the above discussions, we are now ready to model the dynamics of the problem, which is defined by the change in the system state with respect to time. To do this, we track the number of beds occupied in a hospital *p* and an in situ hospital *k* as follows: (6)$$\begin{array}{ll} H_{t+1}^{p}=H_{t}^{p}\left(1-\gamma^{p}\right)+\sum_{q\in \Psi }\sum_{i\in I}y_{i,t}^{q,p}+\sum_{i\in I}x_{i,t}^{p}-\sum_{q\in \Psi }\sum_{i\in I}y_{i,t}^{p,q}-\sum_{k\in \Omega }\sum_{i\in I}z_{i,t}^{p,k}, & \;\forall p\in \Psi ,\forall t\in T, \end{array}$$(7)$$\begin{array}{ll} H_{t+1}^{k}=H_{t}^{k}\left(1-\gamma^{k}\right)+\sum_{i\in I}{\bar{x}}_{i,t}^{k}+\sum_{p\in \Psi }\sum_{i\in I}z_{i,t}^{p,k}, & \;\forall k\in \Omega ,\forall t\in T, \end{array}$$
where $\gamma^{p}$ and $\gamma^{k}$ represent the respective recovery rates of patients in hospital *p* and in situ hospital *k*. Since hospitals vary in their treatment capabilities, recovery rates may differ accordingly. In what follows we explain the details of Equation (Equation 6) for a hospital *p*, where a similar illustration can be argued for Equation (7) for an in situ hospital *k*. To that end, the right-hand side $H_{t+1}^{p}$ is the number of occupied beds at the next point in time t+1. This is based on however many beds were occupied at time *t*, i.e., $H_{t}^{p}$, minus those who recovered, i.e., $\gamma^{p}H_{t}^{p}$, and those who got transferred to another hospital and in situ hospitals, i.e., $\sum_{q\in \Psi }\sum_{i\in I}y_{i,t}^{p,q}+\sum_{k\in \Omega }\sum_{i\in I}z_{i,t}^{p,k}$, plus those who were transferred to *p*, i.e., $\sum_{q\in \Psi }\sum_{i\in I}y_{i,t}^{q,p}$, and patients who were admitted at time *t*, i.e., $\sum_{i\in I}x_{i,t}^{p}$.

In Figure 1, we present an illustrative example of the underlying mechanism of the problem at time *t* for patients of type *i*. In this example, we consider a pandemic as a source node that is flowing patients into the healthcare system. The number of patients of type *i* is denoted as $d_{i,t}$. The node Policy makers represents healthcare agencies who are responsible for admitting patients into the system and distributing them to the hospitals and in situ hosptials. In this toy problem, we are only showing two hospitals, H1 and H2, and two in situ hospitals, H3 and H4. As such, we denote by $x_{i,t}^{1}$, $x_{i,t}^{2}$, ${\bar{x}}_{i,t}^{3}$, and ${\bar{x}}_{i,t}^{4}$ the number of patients assigned to H1, H2, and in situ hospitals H3 and H4 (resp.). Hospitals, on the other hand, can make decisions to redistribute patients and transfer them between each other and to in situ hospitals with variables $y_{i,t}^{1,2}$, $y_{i,t}^{2,1}$, $z_{i,t}^{1,3}$, $z_{i,t}^{1,4}$, $z_{i,t}^{2,3}$, and $z_{i,t}^{2,4}$. Note that in situ hospitals can only admit patients if they are open and operating. This is represented with the variables $u_{t}^{3}$ and $u_{t}^{4}$. Buffer beds (non-pandemic beds) *B* are to be split over hospitals with variables $b_{t}^{1}$ and $b_{t}^{2}$. Each hospital and in situ hospital can track their number of occupied beds with a state variable denoted as $H_{t}^{1}$, $H_{t}^{2}$, $H_{t}^{3}$, and $H_{t}^{4}$. Now that we have defined the decision variables in this toy problem, we now move towards defining the constraints of the problem. To that end, Equation (Equation 3) presents two constraints: the one on the left guarantees that buffer beds borrowed by H1 and H2 cannot exceed *B*. The one on the right states that the number of occupied beds in H1 and H2 cannot exceed the capacity of H1 and H2, i.e., $Q^{1}+b_{t}^{1}$ and $Q^{2}+b_{t}^{2}$. Equation (Equation 4) also presents two constraints: the one on the left basically states that the total number of tents needed to operate the opened in situ hospitals H3 and H4 has to be less than the available number of tents *O*. The one on the right guarantees that, if H3 and H4 are opened, the number of occupied beds in H3 and H4 cannot exceed the capacity of H3 and H4, i.e., $Q^{3}$ and $Q^{4}$. Equation (Equation 5) provides a constraint guaranteeing that all transferred patients out of H1 and H2, i.e., $y_{i,t}^{1,2}+z_{i,t}^{1,3}+z_{i,t}^{1,4}$, and $y_{i,t}^{2,1}+z_{i,t}^{2,3}+z_{i,t}^{2,4}$, have to be less than the number of patients who are in H1 and H2, i.e., $H_{t}^{1}$ and $H_{t}^{2}$ at time *t*. Equations (Equation 6) and (7) respectively allow for tracking the number of occupied beds at each t∈T. To clarify more, we provide an explanation of H1 only. The right-hand side $H_{t+1}^{1}$ is calculated based on however many beds were occupied at time *t*, i.e., $H_{t}^{1}$, minus those who recovered, i.e., $\gamma^{1}H_{t}^{1}$, and those who got transferred to H2, H3, and H4, i.e., $y_{i,t}^{1,2}+z_{i,t}^{1,3}+z_{i,t}^{1,4}$, plus those who were transferred to H1, i.e., $y_{i,t}^{2,1}$, and patients who were admitted at time *t* and assigned to H1, i.e., $\sum_{i\in I}x_{i,t}^{1}$. Similarly for an in situ hospital H3, the right-hand side $H_{t+1}^{3}$ is calculated based on however many beds were occupied at time *t*, i.e., $H_{t}^{3}$, minus those who recovered, i.e., $\gamma^{3}H_{3}^{1}$, plus those who were transferred to H3, i.e., $z_{i,t}^{1,3}+z_{i,t}^{2,3}$, and patients who were admitted at time *t* and assigned to H3, i.e., $\sum_{i\in I}x_{i,t}^{3}$. Finally, the objective of this problem is to find an optimal plan of bed allocation for patients that minimizes the total healthcare costs, which consists of the following: (i) cost of occupying beds in H1, H2, H3, and H4 with a cost of $\sum_{t\in T}\left(W_{t}^{1}H_{t}^{1}+W_{t}^{1}H_{t}^{2}\right)+\sum_{t\in T}\left(W_{t}^{3}H_{t}^{3}+W_{t}^{4}H_{t}^{4}\right)$. (ii) cost of transferring patients, i.e., $\sum_{t\in T}\sum_{i\in I}g^{1,2}y_{i,t}^{1,2}+g^{1,3}z_{i,t}^{1,3}+g^{1,4}z_{i,t}^{1,4}+g^{2,1}y_{i,t}^{2,1}+g^{2,3}z_{i,t}^{2,3}+g^{2,4}z_{i,t}^{2,4}$. (iii) cost of rejecting patients, i.e., $\sum_{t\in T}\sum_{i\in I}c_{i,t}\left\{\max\left\{d_{i,t}-x_{i,t}^{1}-x_{i,t}^{2}-{\bar{x}}_{i,t}^{3}-{\bar{x}}_{i,t}^{4},0\right\}\right\}$.

In this paper, we aim to minimize the total healthcare costs. This includes the cost of shortage, which reflects the *patient rejection cost*, hospital and in situ hospital operating costs, and cost of patient transport, which reflects *logistical costs*. More formally, we write our objective function *f* as follows: (8)$$\begin{array}{cc} f & =\sum_{t\in T}\sum_{p\in \Psi }W_{t}^{p}H_{t}^{p}+\sum_{t\in T}\sum_{k\in \Omega }W_{t}^{k}H_{t}^{k}+\sum_{t\in T}\sum_{i\in I}\sum_{p\in \Psi }\sum_{q\in \Psi }g^{p,q}y_{i,t}^{p,q}+\sum_{t\in T}\sum_{i\in I}\sum_{p\in \Psi }\sum_{k\in \Omega }g^{p,k}z_{i,t}^{p,k}\\ \; & +\sum_{t\in T}\sum_{i\in I}c_{i,t}\left\{\max\left\{d_{i,t}-\sum_{p\in \Psi }x_{i,t}^{p}-\sum_{k\in \Omega }{\bar{x}}_{i,t}^{k},0\right\}\right\}. \end{array}$$The first two terms correspond to the total per-patient operating costs for both hospitals and in situ hospitals. The second two terms are associated with the patient transfer cost. Finally, the last term is the cost of demand shortage. This makes *f* a multi-criterion objective function. Clearly, there is a trade-off: as logistical expenses increase, the cost of demand shortage decreases and vice versa. This underscores the need for an optimization-based model, which we present in the next section, to effectively balance this trade-off. We refer readers to Table A1 in Appendix A for a comprehensive summary of the paper’s notation.

### 2.1. Multi-Period Optimization Problem (Equation 9)

With the aforementioned discussions, we are now ready to present our optimization problem in what follows:(9)$$\begin{array}{cccc} F^{*}\equiv \mathrm{minimize} & \;f=\sum_{t\in T}\sum_{p\in \Psi }W_{t}^{p}H_{t}^{p}+\sum_{t\in T}\sum_{k\in \Omega }W_{t}^{k}H_{t}^{k}\\ \; & \;+\sum_{t\in T}\sum_{i\in I}\sum_{p\in \Psi }\sum_{q\in \Psi }g^{p,q}y_{i,t}^{p,q}+\sum_{t\in T}\sum_{i\in I}\sum_{p\in \Psi }\sum_{k\in \Omega }g^{p,k}z_{i,t}^{p,k}\\ \; & \;+\sum_{t\in T}\sum_{i\in I}c_{i,t}\max\left\{d_{i,t}-\sum_{p\in \Psi }x_{i,t}^{p}-\sum_{k\in \Omega }{\bar{x}}_{i,t}^{k},0\right\}\\ \mathrm{subject}\;\mathrm{to} & \;H_{t+1}^{p}=H_{t}^{p}\left(1-\gamma^{p}\right)+\sum_{q\in \Psi }\sum_{i\in I}y_{i,t}^{q,p}+\sum_{i\in I}x_{i,t}^{p}-\sum_{q\in \Psi }\sum_{i\in I}y_{i,t}^{p,q}-\sum_{k\in \Omega }\sum_{i\in I}z_{i,t}^{p,k}, & \; & \;\forall p\in \Psi ,\forall t\in T,\\ \; & \;\sum_{q\in \Psi }\sum_{i\in I}y_{i,t}^{p,q}+\sum_{k\in \Omega }\sum_{i\in I}z_{i,t}^{p,k}\leq H_{t}^{p}, & \; & \;\forall p\in \Psi ,\forall t\in T,\\ \; & \;H_{t}^{p}\leq Q^{p}+b_{t}^{p}, & \; & \;\forall p\in \Psi ,\forall t\in T,\\ \; & \;\sum_{p\in \Psi }b_{t}^{p}\leq B, & \; & \;\forall t\in T,\\ \; & \;H_{t}^{k}\leq u_{t}^{k}{\bar{Q}}^{k}, & \; & \;\forall k\in \Omega ,\forall t\in T,\\ \; & \;H_{t+1}^{k}=H_{t}^{k}\left(1-\gamma^{k}\right)+\sum_{i\in I}{\bar{x}}_{i,t}^{k}+\sum_{p\in \Psi }\sum_{i\in I}z_{i,t}^{p,k}, & \; & \;\forall k\in \Omega ,\forall t\in T,\\ \; & \;\sum_{k\in \Omega }o^{k}u_{t}^{k}\leq O, & \; & \;\forall t\in T,\\ \; & \;u_{t}^{k}\in \left\{0,1\right\},\;H_{t}^{p},H_{t}^{k},y_{i,t}^{p,q},x_{i,t}^{p},{\bar{x}}_{i,t}^{k},z_{i,t}^{p,k}\geq 0,\\ & \;\forall i\in I,\forall p,q\in \Psi ,k\in \Omega ,\forall t\in T. \end{array}$$We note that in Appendix C, we present a modeling extension that allows hospitals to independently allocate beds between pandemic and non-pandemic cases, along with the associated results. This approach enables hospitals to better manage bed availability based on patient health needs. Problem (Equation 9) is a multi-period scheduling problem designed to optimally accommodate patient demand during a pandemic while operating under resource constraints. In our problem, we consider different resources, which include bed capacities of hospitals and in situ hospitals $Q^{p}$ and $Q^{k}$, buffer beds *B*, and tents *O*. The optimal solution of Problem (Equation 9) provides a complete set of information required to optimally accommodate patient demand. Specifically, at every point in time, healthcare policy agents would like to know how many patients of different severity levels enter the healthcare system ($x_{i,t}^{p}$ and ${\bar{x}}_{i,t}^{k}$). Hospitals and in situ hospitals (if opened and operated $u_{t}^{k}$) would track the bed utilization ($H_{t}^{p}$ and $H_{t}^{k}$) and would like to know how many non-pandemic beds should be used ($b_{t}^{p}$) and how many patients can be transferred ($y_{i,t}^{p,q}$ and $z_{i,t}^{p,k}$). As such, solving this optimization model enables an efficient allocation of these limited resources, ultimately minimizing total healthcare costs. We begin with the following assumptions.

**Assumption 1.** 

*Assume that $c_{i,t}>0$ for some (i,t)∈I×T.*


Suppose that the cost of shortage is 0, that is, $c_{i,t}=0$ for all i∈I and t∈T. Then, in this case, no patients should be hospitalized and no one should enter the system, that is, $x_{i,t}^{p}={\bar{x}}_{i,t}^{k}=0$. Consequently, with the initial state conditions, all decision and state variables are trivially set to 0, which means that $F^{*}=0$. Thus, Assumption 1 enriches the problem and highlights its importance for further study. Next, we turn our attention towards the computational aspects of Problem (Equation 9).

#### 2.1.1. Computational Aspects of Problem (Equation 9)

The decision variables $x_{i,t}^{p}$, ${\bar{x}}_{i,t}^{k}$, $H_{t}^{p}$, $H_{t}^{k}$, $b_{t}^{p}$, $y_{i,t}^{p,q}$, and $z_{i,t}^{p,k}$ should theoretically be integers. However, given that this problem deals with large populations, we can safely relax the integrality constraints of these variables. Practically speaking, this implies that the solutions of the relaxed and integer models are sufficiently close. The major benefit of this is that it can drastically reduce the computational complexity of Problem (Equation 9). Toward that end, in Table 2, we explicitly describe the computational aspects of problem (Equation 9).

Notably, as the problem size grows, the number of decision variables and constraints increases polynomially. To illustrate a realistic scenario, we consider the state of Virginia (VA), USA, and report the size of the largest variable, $y_{i,t}^{p,q}$. The number of hospitals in VA is 99 [37]. As such, if the considered time horizon is 7 days and the count of the severity levels is 3, this means the size of $y_{i,t}^{p,q}$ is 205,821, which in turn means that Problem (Equation 9) is computationally expensive to solve. Toward that end, in the next section, we propose a more efficient reformulation that allows us to drastically reduce the computational complexity.

#### 2.1.2. An Equivalent Reformulation of (Equation 9)

In this section, we present an equivalent reformulation of Problem (Equation 9) under a mild condition. We then present a theoretical result that demonstrates the equivalency of both optimization models. Then, we present a numerical analysis that shows the computational gains of the newly proposed reformulation. We present our reformulation in what is next.

In real-world scenarios, the primary cost associated with transferring patients between hospitals and in situ facilities is the fixed expense of procuring ambulances, while the actual transportation cost is relatively negligible. Therefore, without loss of generality, the following assumption can be made.

**Assumption 2.** 

*Assume that $g^{p,q}\approx g$ and $g^{p,k}\approx g$, where g refers to the fixed cost (e.g., ambulance utilization).*


In what follows, we rely on the above assumption to construct an equivalent reformulation of (Equation 9). The key motive behind this reformulation is to simplify the associated computational aspects. Towards that end, we start by defining a new node 0 as a dummy node for exchanging patients between hospitals and in situ hospitals. As such, one can lump the number of patients moving out from a hospital *p* and transferred to all other hospitals with a decision variable $y_{i,t}^{p,0}$ and lump the number of patients moving into a hospital *p* from all other hospitals with $y_{i,t}^{0,p}$. This allows to write the following equations:(10)$$\begin{array}{c} \sum_{q\in \Psi }y_{i,t}^{p,q}=y_{i,t}^{p,0}\;\mathrm{and}\;\sum_{q\in \Psi }y_{i,t}^{q,p}=y_{i,t}^{0,p},\;\forall p\in \Psi ,\forall t\in T,\forall i\in I. \end{array}$$

In a similar fashion, we lump the number of patients moving out from any hospital *p* to all destination in situ hospitals, with the decision variable $z_{i,t}^{p,0}$, and the number of patients moving into any in situ hospital *k*, with $z_{i,t}^{0,k}$. This can be formally expressed as(11)$$\begin{array}{cc} \; & \sum_{k\in \Omega }z_{i,t}^{p,k}=z_{i,t}^{p,0}\;\mathrm{and}\;\sum_{p\in \Psi }z_{i,t}^{p,k}=z_{i,t}^{0,k},\;\forall k\in \Omega ,\forall t\in T,\forall i\in I. \end{array}$$Introducing the dummy node 0 necessitates governing flow-balance constraints between patients moving in and out of node 0. This relation is derived as(12)$$\begin{array}{ll} \sum_{i\in I}\sum_{p\in \Psi }\sum_{q\in \Psi }y_{i,t}^{p,q}+\sum_{i\in I}\sum_{p\in \Psi }\sum_{k\in \Omega }z_{i,t}^{p,k}=\sum_{i\in I}\sum_{p\in \Psi }\sum_{q\in \Psi }y_{i,t}^{p,q}+\sum_{i\in I}\sum_{p\in \Psi }\sum_{k\in \Omega }z_{i,t}^{p,k}, & \forall t\in T, \end{array}$$(13)$$\begin{array}{ll} \sum_{i\in I}\sum_{p\in \Psi }y_{i,t}^{p,0}+\sum_{i\in I}\sum_{p\in \Psi }z_{i,t}^{p,0}=\sum_{i\in I}\sum_{p\in \Psi }y_{i,t}^{0,p}+\sum_{i\in I}\sum_{k\in \Omega }z_{i,t}^{0,k}, & \;\forall t\in T, \end{array}$$
where Equality (Equation 12) is trivial. As can be seen, Equation (13) guarantees that for each time *t* the total number of patients exchanged is conserved. The aforementioned definitions allows us to introduce our proposed reformulation as follows.(14)$$\begin{array}{cccc} F^{\mathrm{equiv}}\equiv \mathrm{minimize} & \;f^{\mathrm{equiv}}=\sum_{t\in T}\sum_{k\in \Omega }W_{t}^{k}H_{t}^{k}+\sum_{t\in T}\sum_{p\in \Psi }W_{t}^{p}H_{t}^{p}\\ \; & \;+\frac{1}{2}g\sum_{t\in T}\sum_{i\in I}\sum_{p\in \Psi }\left(y_{i,t}^{p,0}+y_{i,t}^{0,p}\right)+\frac{1}{2}g\sum_{t\in T}\sum_{i\in I}\sum_{p\in \Psi }z_{i,t}^{p,0}+\frac{1}{2}g\sum_{t\in T}\sum_{i\in I}\sum_{k\in \Omega }z_{i,t}^{0,k}\\ \; & \;+\sum_{t\in T}\sum_{i\in I}c_{i,t}\max\left\{d_{i,t}-\sum_{p\in \Psi }x_{i,t}^{p}-\sum_{k\in \Omega }{\bar{x}}_{i,t}^{k},0\right\}\\ \mathrm{subject}\;\mathrm{to} & \;H_{t+1}^{p}=H_{t}^{p}\left(1-\gamma^{p}\right)+\sum_{i\in I}y_{i,t}^{0,p}+\sum_{i\in I}x_{i,t}^{p}-\sum_{i\in I}y_{i,t}^{p,0}-\sum_{i\in I}z_{i,t}^{p,0}, & \; & \;\forall p\in \Psi ,\forall t\in T\\ \; & \;\sum_{i\in I}y_{i,t}^{p,0}+\sum_{i\in I}z_{i,t}^{p,0}\leq H_{t}^{p}, & \; & \;\forall p\in \Psi ,\forall t\in T\\ \; & \;H_{t}^{p}\leq Q^{p}+b_{t}^{p}, & \; & \;\forall p\in \Psi ,\forall t\in T\\ \; & \;\sum_{p\in \Psi }b_{t}^{p}\leq B, & \; & \;\forall t\in T\\ \; & \;\sum_{i\in I}\sum_{p\in \Psi }y_{i,t}^{p,0}+\sum_{i\in I}\sum_{p\in \Psi }z_{i,t}^{p,0}\leq \sum_{i\in I}\sum_{p\in \Psi }y_{i,t}^{0,p}+\sum_{i\in I}\sum_{k\in \Omega }z_{i,t}^{0,k}, & \; & \;\forall t\in T\\ \; & \;H_{t}^{k}\leq u_{t}^{k}{\bar{Q}}^{k}, & \; & \;\forall k\in \Omega ,\forall t\in T\\ \; & \;H_{t+1}^{k}=H_{t}^{k}\left(1-\gamma^{k}\right)+\sum_{i\in I}{\bar{x}}_{i,t}^{k}+\sum_{i\in I}z_{i,t}^{0,k}, & \; & \;\forall k\in \Omega ,\forall t\in T\\ \; & \;\sum_{k\in \Omega }o^{k}u_{t}^{k}\leq O, & \; & \;\forall t\in T,\\ \; & \;u_{t}^{k}\in \left\{0,1\right\},\;H_{t}^{p},H_{t}^{k},y_{i,t}^{p,0},y_{i,t}^{0,p},x_{i,t}^{p},{\bar{x}}_{i,t}^{k},z_{i,t}^{p,0},z_{i,t}^{0,k}\geq 0,\\ & \;\forall i\in I,\forall p\in \Psi ,k\in \Omega ,\forall t\in T. \end{array}$$

Note that Constraint (13) was further simplified by replacing the equality with an inequality (for more details, the readers are referred to the proof of Proposition 1 in Appendix B). We next present our first result in Proposition 1, which simply states that both optimization problems are equivalent.

**Proposition 1.** 

*If the conditions in Assumption 2 are satisfied, then Problem (Equation 14) is equivalent to Problem (Equation 9), with $F^{\mathrm{equiv}}=F^{*}$.*


The main premise in the proposed reformulation is to reduce the problem size and consequently the computational complexity. As such, we present in Figure 2 the % improvement in the runtime (in minutes) of Problem (Equation 14) over Problem (Equation 9). We compare the performance on two crucial parts, which are the data processing, i.e., building the optimization model components, and the optimization runtime itself. Eventually, both will be used when the proposed framework is implemented. Gurobi solver version 12.0.0 [38] was considered for solving the optimization models. To that end, we vary the problem size, measured by the number of hospitals, which mainly controls the complexity.

As noticed, as the problem size increases, (Equation 14) offers drastic improvements, ranging between 60% and 98%, in the runtime of both the optimization and the data processing, which hence favors it computationally over (Equation 9).

### 2.2. Structural Properties of (Equation 14)

In this section, we present structural properties of (Equation 14) which provide additional insights into the optimal solution under some conditions. We begin by assuming that $\sum_{k}o^{k}\leq O$, meaning that all in situ hospitals can be operated if necessary. This allows us to define $D_{t}^{\mathrm{sys}}$ and $S_{t}^{\mathrm{sys}}$ as the respective system demand and supply of hospital beds at time *t*, which can be written as follows:$$D_{t}^{\mathrm{sys}}=\sum_{i\in I}d_{i,t},\;S_{t}^{\mathrm{sys}}=\sum_{p\in \Psi }\left(Q^{p}-H_{t}^{p}\right)+B+\sum_{k\in \Omega }\left({\bar{Q}}^{k}-H_{t}^{k}\right)\;\forall t\in T,$$
where $Q^{p}-H_{t}^{p}$ and ${\bar{Q}}^{k}-H_{t}^{k}$ represent the available beds, i.e., unoccupied, for a hospital *p* and in situ hospital *k* at time *t*. If the $D_{t}^{\mathrm{sys}}\leq S_{t}^{\mathrm{sys}},\forall t$, the healthcare system can accommodate all patients throughout the pandemic, minimizing overall patient rejection costs. Conversely, if $D_{t}^{\mathrm{sys}}\geq S_{t}^{\mathrm{sys}},\forall t$, policymakers must address the shortfall by prioritizing hospitalizations based on factors such as patient health status, hospitalization costs, and other relevant considerations. Next, we present a theorem that defines the optimal solution under each case. To simplify the exposure of the mathematical equations in the following theorem, we assume that |I|=1, which means we can drop the subscript *i*.

**Theorem 1.** 

*Suppose that*

(15)
$$\begin{array}{cc} \; & \sum_{t\in T}\sum_{k\in \Omega }W_{t}^{k}{\bar{Q}}^{k}+\sum_{t\in T}\sum_{p\in \Psi }W_{t}^{p}\left(Q^{p}+B\right)+\frac{1}{2}g\sum_{t\in T}\sum_{p\in \Psi }\left(2Q^{p}+2B\right)\\ \; & +\frac{1}{2}g\sum_{t\in T}\sum_{i\in I}\sum_{p\in \Psi }\left(Q^{p}+B\right)+\frac{1}{2}g\sum_{t\in T}\sum_{k\in \Omega }{\bar{Q}}^{k}\leq \min_{t}c_{t} \end{array}$$

*If $c_{t}=C,\;\forall t\in T$, then the following two results hold:*

*If $D_{t}^{\mathrm{sys}}\geq S_{t}^{\mathrm{sys}},\forall t$, then the optimal solution satisfy the following,*

(16)
$$\begin{array}{cccc} \; & H_{t}^{p}=Q^{p}+\xi^{p}B, & \; & \forall p\in \Psi ,\forall t\in T,\\ \; & H_{t+1}^{p}=\left(1-\gamma^{p}\right)H_{t}^{p}+x_{t}^{p}, & \; & \forall p\in \Psi ,\forall t\in T,\\ \; & b_{t}^{p}=\xi^{p}B, & \; & \forall p\in \Psi ,\forall t\in T,\\ \; & u_{t}^{k}=1, & \; & \forall k\in \Omega ,\forall t\in T,\\ \; & H_{t}^{k}={\bar{Q}}^{k}, & \; & \forall k\in \Omega ,\forall t\in T,\\ \; & H_{t+1}^{k}=\left(1-\gamma^{k}\right)H_{t}^{k}+x_{t}^{k}, & \; & \forall k\in \Omega ,\forall t\in T,\\ & y_{t}^{p,0}=y_{t}^{0,p}=z_{t}^{p,0}=z_{t}^{0,k}=0, & \; & \forall p\in \Psi ,\forall k\in \Omega ,\forall t\in T,\\ \; & \sum_{p\in \Psi }x_{t}^{p}+\sum_{k\in \Omega }{\bar{x}}_{t}^{k}=\sum_{p\in \Psi }Q^{p}+B+\sum_{k\in \Omega }{\bar{Q}}^{k}, & \; & \forall t\in T,\\ \; & \xi^{p}\geq 0,\;H_{0}^{p}=0\;\forall p\in \Psi ,H_{0}^{k}=0\;\forall k\in \Omega ,\;\sum_{p\in \Psi }\xi^{p}=1. \end{array}$$


*If $D_{t}^{\mathrm{sys}}\leq S_{t}^{\mathrm{sys}},\forall t$, and if $W_{t}^{p}=W^{p},\forall p\in \Psi$ and $W_{t}^{k}=W^{k},\forall k\in \Omega$, for all t, we have the following. Define $J=\left\{1\leq j\leq \left|\Psi \right|+|\Omega |,\;W^{j}\leq W^{j+1},\;j\in \Psi \cup \Omega \right\}$, as a set of hospitals and in situ hospitals ranked based on their operating costs. Then the optimal solution satisfies the following (we assume, without loss of generality, that one of the hospitals has the lowest operating cost among all hospitals and in situ hospitals),*

(17)
$$\begin{array}{cccc} \; & x_{t}^{1}=\min\{\max\{0,d_{t}-B-\left(Q^{1}-\left(1-\gamma^{1}\right)H_{t}^{1}\right)\},Q^{1}+B-\left(1-\gamma^{1}\right)H_{t}^{1}\}, & \; & \forall t\in T,\\ \; & b_{t}^{1}=B, & \; & \forall t\in T,\\ \; & x_{t}^{j}=\min\{\max\{0,d_{t}-B-\sum_{\begin{array}{c} k=1\\ k\in J \end{array}}^{j-1}\left(Q^{k}-\left(1-\gamma^{k}\right)H_{t}^{k}\right)\},Q^{j}-\left(1-\gamma^{k}\right)H_{t}^{k}\}, & \; & \forall j\in J,\forall t\in T,\\ \; & H_{t+1}^{j}=\left(1-\gamma^{j}\right)H_{t}^{j}+x_{t}^{j}, & \; & \forall j\in J,\forall t\in T,\\ \; & u_{t}^{k}=1, & \; & \forall k\in \Omega ,\forall t\in T,\\ \; & y_{t}^{p,0}=y_{t}^{0,p}=z_{t}^{p,0}=z_{t}^{0,k}=0, & \; & \forall p\in \Psi ,\forall k\in \Omega ,\forall t\in T,\\ \; & \sum_{p\in \Psi }x_{t}^{p}+\sum_{k\in \Omega }{\bar{x}}_{t}^{k}=\sum_{i\in I}d_{i,t}, & \; & \forall t\in T, \end{array}$$




Condition (Equation 15) in Theorem 1 is not too unrealistic. During peak infection periods, the cost of rejecting a patient in need of hospitalization may outweigh other logistical expenses, as such rejections could result in catastrophic outcomes (e.g., patient death). To that end, the results presented in Theorem 1 are insightful and intuitive: First, when the patient demand exceeds the available hospital bed capacity, the optimal strategy is to maximize bed utilization. This entails fully occupying all hospital and in situ hospital beds while strategically distributing buffer beds across hospitals to accommodate the excess demand and minimize deaths. Second, when patient demand is lower than the available hospital bed supply, the optimal strategy is to fully accommodate the demand while allocating patients efficiently. This is done greedily, prioritizing hospitals and in situ facilities with the lowest operating costs first, then progressively utilizing higher-cost options until all demand is met.

### 2.3. A Robust Reformulation

Healthcare agencies often depend on well-established models to forecast disease progression within a population over time, as these methods have demonstrated reliable predictive accuracy. Among the most widely used are compartmental models such as the SIR model and its various extensions. However, in practice, the actual number of infections can significantly diverge from these predictions. Such discrepancies—particularly underestimations—can lead to serious consequences. When predicted infection rates fall short of actual figures, the healthcare system may face a shortage of allocated pandemic beds. This mismatch can result in patient overflow, denial of critical care, and ultimately, increased healthcare costs. Therefore, robust strategies are essential to mitigate the impact of such uncertainties, ensuring adequate preparedness and minimizing the risk of resource shortages.

To that end, in this section, we extend our formulation by considering that the patient demand is $d_{i,t}$ is uncertain. To model uncertainty, we consider a robust approach, which means that we assume that the demand $d_{i,t}\in D_{i,t}$, where $D_{i,t}\equiv \left[d_{i,t}^{l},d_{i,t}^{u}\right]$ represents a time-dependent confidence interval, with lower and upper bounds denoted as $d_{i,t}^{l}$ and $d_{i,t}^{u}$. In this paper, we adopt a robust approach. This means that the corresponding formulation is the same as that of (Equation 14); however the objective function, now denoted as $F^{\mathrm{robust}}$, can be written as(18)$$\begin{array}{l} F^{\mathrm{robust}}\equiv \mathrm{minimize}\max_{d_{i,t}\in D_{i,t}}\sum_{t\in T}\sum_{k\in \Omega }W_{t}^{k}H_{t}^{k}+\sum_{t\in T}\sum_{p\in \Psi }W_{t}^{p}H_{t}^{p} \end{array}$$(19)$$\begin{array}{l} \;+\frac{1}{2}g\sum_{t\in T}\sum_{i\in I}\sum_{p\in \Psi }\left(y_{i,t}^{p,0}+y_{i,t}^{0,p}\right)+\frac{1}{2}g\sum_{t\in T}\sum_{i\in I}\sum_{p\in \Psi }z_{i,t}^{p,0}+\frac{1}{2}g\sum_{t\in T}\sum_{i\in I}\sum_{k\in \Omega }z_{i,t}^{0,k} \end{array}$$(20)$$\begin{array}{l} +\sum_{t\in T}\sum_{i\in I}c_{i,t}\{\max\{d_{i,t}-\sum_{p\in \Psi }x_{i,t}^{p}-\sum_{k\in \Omega }{\bar{x}}_{i,t}^{k},0\}\} \end{array}$$

In what follows, we present a proposition that ensures that solving (Equation 14) under uncertain demand is equivalent to solving the deterministic counterpart simply by setting $d_{i,t}$ to its upper bound $d_{i,t}^{u}$ for all i∈I and t∈T.

**Proposition 2.** 

*The optimal solution $X_{\mathrm{robust}}^{*}$ to the robust optimization problem is the solution to the deterministic counterpart, i.e., (Equation 14), by setting $d_{i,t}=d_{i,t}^{u}\;\forall i\in I,\forall t\in T$.*


The above proposition simply states that the robust optimization problem has the same computational complexity as the deterministic counterpart. This result, while being simple, its significant as it allows for generating solutions that are robust to demand perturbations.

### 2.4. Simulating Demand on Hospital Beds

To estimate the potential demand arising from future disease outbreaks, we employ a compartmental SEIRD model—one of the widely accepted frameworks in the epidemiological literature. The SEIRD model partitions a total population of size *N* into five mutually exclusive compartments: Susceptible (S), Exposed (E), Infected (I), Recovered (R), and Deceased (D). These compartments respectively represent the number of individuals who are vulnerable to infection, have been exposed but are not yet infectious, are currently infected, have recovered and acquired immunity, and have succumbed to the disease, all as functions of time *t* in a time horizon T (see, e.g., [39]). The mathematical model is written as follows:(21)$$\begin{array}{l} S_{t+1}=S_{t}-\frac{\beta S_{t}I_{t}}{N}+\alpha E_{t}+\tau R_{t}, \end{array}$$(22)$$\begin{array}{l} E_{t+1}=E_{t}+\frac{\beta S_{t}I_{t}}{N}-\rho E_{t}-\alpha E_{t}, \end{array}$$(23)$$\begin{array}{l} I_{t+1}=I_{t}+\rho E_{t}-\left(\gamma +\delta \right)I_{t}, \end{array}$$(24)$$\begin{array}{l} R_{t+1}=R_{t}+\gamma I_{t}-\tau R_{t}. \end{array}$$(25)$$\begin{array}{l} D_{t+1}=D_{t}+\delta I_{t}, \end{array}$$Parameter β denotes the rate of exposure to infected individuals. α and ρ denote the probability and the rate of incubated individuals turning back to susceptible and to infectious, respectively. γ and δ denote the rates of recovery and mortality. To better model the disease dynamics, we consider, in addition to the basic SEIRD model introduced by [39,40,41], the possibility of reinfections at a rate τ, in which recovered individuals can rejoin class S losing immunity which for COVID-19 lasts for up to 4 months [42].

Given the number of infections $I_{t}$, the corresponding demand $d_{i,t}$ can be derived using Equations (Equation 1) and (Equation 2). However, as discussed in Section 2.3, demand is subject to uncertainty and is assumed to lie within the uncertainty set $D_{i,t}$. In the following, we elaborate on the construction of these uncertainty sets, leveraging the SEIRD model as the underlying epidemiological framework.

To that end, note that one of the most critical parameters in the SEIRD model governing disease transmission—namely, β—can be subject to uncertainty. This uncertainty is often driven by variations in population compliance with lockdown measures. Factors such as funeral gatherings, children engaging in outdoor activities, high attendance at schools and universities, and unjustified in-person meetings can all cause the actual interaction rate to deviate significantly from its intended level. This simply means that varying $\beta \in [\beta^{l},\beta^{u}]$ will result in different realizations of $I_{t}$ and hence $d_{i,t}$. We define $\beta^{l}$ and $\beta^{u}$ as follows:(26)$$\beta^{l}=\beta^{\mathrm{avg}}\left(1-r\right),\;\beta^{u}=\beta^{\mathrm{avg}}\left(1+r\right),$$
where $\beta^{\mathrm{avg}}$ represents an average transmission rate, which can be estimated using historical data on interaction rates across different time periods. Parameter *r*, on the other hand, defines the confidence level in which the mean estimate $\beta^{\mathrm{avg}}$ lies in $\left[\beta^{l},\beta^{u}\right]$. In addition, it can also be interpreted as a reflection of the decision-maker’s level of conservatism.

## 3. Case Study: COVID-19 Pandemic and Bed Scheduling in Northern Virginia (NOVA)

### 3.1. Case Study Setup

In order to show the effectiveness of our framework, we consider a case study that utilizes the data for the COVID-19 outbreak in the NOVA area. By September 2021, hospitals in NOVA were already approaching capacity due to the rise in COVID-19 cases and faced additional strain with the arrival of Afghan refugees. According to the Virginia Hospital and Healthcare Association, hospitalizations increased by 1008% and new cases rose by 1217% over a two-month period, resulting in severely overcrowded emergency rooms and widespread staffing shortages across the state [16]. To that end, we consider the weekly number of infections reported by the Centers for Disease Control and Prevention (CDC) from 2 October 2021 to 26 February 2022 in NOVA [43]. Recall that the number of infections $I_{t}$ is translated into patient demand $d_{i,t}$ in our model (using (Equation 1) and (Equation 2)). In Figure 3, we show the weekly patient demand in NOVA. As is noticeable, the demand is split into two categories: mild and severe symptoms, i.e., |I|=2. Given that the CDC reports weekly deaths for a given state, we first extract the weekly deaths in the state of Virginia and then estimate the weekly infections, considering that the mortality probability is 2% [44]. To limit the analysis to the NOVA area, we multiply the weekly infections by 30% (this is based on the fact that the proportion of the NOVA population in Virginia is 30%) [45]. We then split the number of infections into mild and severe symptoms using the proportions 80% and 20% (resp.) [46]. From the perspective of our model, the key distinction between subjects experiencing mild versus severe symptoms is that mildly symptomatic individuals require hospitalization with standard care beds, whereas those with severe symptoms necessitate intensive care unit (ICU) support and/or mechanical ventilation.

We now present the data collected on existing hospitals and potential in situ hospital locations, including their associated costs and bed capacities. Table 3 provides a summary of the bed capacities for the ten major hospitals considered in NOVA.

In March 2020, several sites in Northern Virginia (NOVA) were identified for potential field hospitals to expand medical capacity in response to the anticipated COVID-19 surge. Planned facilities included the National Conference Center (approximately 1000 beds) in the city of Leesburg, the Dulles Expo Center (at least 500 beds), and George Mason University (at least 500 beds) in the City of Fairfax. Additionally, the Hilton Garden Inn in Neabsco Commons, Woodbridge, was considered for conversion with a potential capacity of 200 beds [57]. We summarize the bed capacity of each of those potential in situ hospitals in Table 4.

We now explain the estimated per-patient weekly costs for hospitals and in situ hospitals, i.e., $W_{t}^{p}$ and $W_{t}^{k}$ (resp.). The cost of hospitalizing a COVID-19 patient depends on illness severity, treatment needs, and length of stay, with median daily costs estimated at approximately USD 2337 in 2020 [58], which means that the weekly cost is simply USD 16,359. Due to limited cost data, we estimate hospital daily costs by ranking facilities based on their technological advancement and ICU capabilities. Key criteria include level of specialized care, affiliation with advanced health systems (e.g., INOVA), and the scale and sophistication of their ICU infrastructure. The middle-listed hospital receives the average cost, i.e., USD 16,359, whereas those with a higher (lower) ranks receive an additional (less) USD 175 in weekly costs in a linear fashion. For the potential in situ hospitals, we consider the same average cost across all in situ hospitals, which is 2 × USD 16,359 = USD 32,718, Costs are reflected in the last column of Table 3 and Table 4. In addition, we note that the per-patient transportation cost is considered to be g=USD940 [59].

The only remaining parameter input to Problem (Equation 14) is (i) the recovery rate $\gamma^{p}=1/10$ for all hospitals and in situ hospitals (due to lack of data, we assume that the recovery rate is the same across all hospitals and in situ hospitals), assuming that the days of stay for a patient is 10 days [60]. (ii) The non-pandemic weekly beds, which are considered to be (the buffer of 898, were mainly estimated from the 68% pandemic hospital bed usage reported by [61]) B=898 [61].

### 3.2. Findings and Design Implications

This section presents the results of our case study analysis for the NOVA region during the COVID-19 pandemic, highlighting both the empirical benefits and operational insights derived from implementing the proposed optimization framework. We evaluate system performance under two distinct strategies: (i) The *current practice (CP)* is characterized by reactive and decentralized decision-making. The CP solution is derived using a greedy algorithm that assigns incoming patients to available beds following the allocation policy outlined in Theorem 1. (ii) The *Optimized Model*, i.e., following (Equation 14), which employs anticipatory, system-wide scheduling to proactively manage capacity and demand.

Figure 4 presents the expected healthcare costs across both approaches, decomposed into rejection costs and logistical costs. The bar on the right represents the total system cost, which is the sum of both costs. The optimized framework achieves a substantial reduction in total cost, decreasing from over USD 2647 billion under CP to approximately USD 1161 billion, representing a savings of more than 56%.

While both approaches face the same underlying demand (Figure 3), the optimized model achieves significantly better outcomes by proactively shaping system behavior. One critical difference lies in how each strategy handles early-stage admissions. CP tends to over-admit mild-symptom patients at the onset of the outbreak, rapidly exhausting bed capacity. This front-loaded saturation forces the system to reject a growing number of severe cases in later weeks, when capacity is most needed. In contrast, the optimization model enforces a carefully calibrated trade-off: it limits mild-case admissions early in order to preserve space for high-acuity patients as demand intensifies. This shift from short-term volume maximization to long-term value optimization enables a more balanced and resilient system response.

To better understand this behavior, Figure 5 shows weekly patient rejections disaggregated by symptom severity. CP admits large volumes of mild cases in early weeks, which contributes directly to the rejection of hundreds of severe patients later on, particularly during the second week of October, when over 355 severe cases are turned away, compared with zero under the optimized model.

The optimized model, by contrast, maintains zero rejections of severe patients across all weeks. This is made possible by deliberately and consistently rejecting a subset of mild-symptom patients, particularly during periods of emerging surge risk. Importantly, this is not a result of conservative planning or rigid rules. Instead, the model embeds a dynamic triage logic in which admissions are guided by the evolving marginal utility of each bed. Every accepted patient is evaluated not only by immediate need but also by the long-term implications for system capacity. This rational prioritization formalizes the type of decision-making that clinicians often carry out under crisis conditions but does so consistently and at scale. In doing so, it reduces preventable mortality while preserving critical care capacity when it matters most.

Another noteworthy insight from Figure 5 is that the CP approach tends to admit all patients, largely ignoring the associated logistical costs. Specifically, it admits more patients with mild symptoms compared with the optimal solution. This is because the optimal approach carefully balances the trade-offs between the costs of treating severe and mild cases, as well as the logistical costs. As a result, the optimal solution prioritizes a higher admission rate for severe cases, assigns lower priority to mild cases, and maintains a moderate level of logistical burden.

We now turn our attention to the role of patient transfers in the optimal solution. Given that hospital costs were fixed over the planning horizon (see Table 3), the optimal strategy recommended no patient transfers. This outcome is intuitive: transferring patients between hospitals becomes advantageous only when future hospital costs are lower than current ones. Such cost variability enables reallocation from higher-cost to lower-cost facilities, effectively preserving capacity in the more economical hospitals for future admissions. To further investigate this dynamic, we conducted a numerical experiment in which we generated 1000 randomized hospital cost profiles that varied over time. The results, presented in Figure 6, illustrate the mean weekly transfer activity across hospitals and in situ facilities, with shaded bands indicating one standard deviation of variability. Interestingly, the mean transfer activity remains largely stable throughout the planning horizon. This consistency reflects an optimal policy that consistently prioritizes freeing capacity at the largest facility—Inova Fairfax Medical Campus—by transferring patients to smaller hospitals and in situ sites. Doing so allows the system to expand capacity and accommodate more patients. This behavior underscores a key aspect of the optimization logic: minimizing patient rejection costs is prioritized over logistical and transfer costs.

Taken together, these results highlight several critical design insights. First, surge planning should treat capacity not as a static resource to be filled but as a dynamic buffer that can absorb uncertainty and mitigate downstream risk. Second, structured prioritization, especially under limited resources, is not only ethically sound but also operationally essential. Third, system-wide coordination and resource sharing can dramatically reduce total healthcare costs, especially when tools like transfers and buffer beds are deployed optimally. Ultimately, the proposed framework offers more than just performance improvements; it provides a replicable strategy for achieving both efficiency and equity under crisis. Rather than treating capacity allocation, patient transfers, and admission decisions as siloed operational problems, the model integrates them into a cohesive and adaptive system-level response. It demonstrates that resilience need not depend on overbuilding or reacting to every surge in real time. Instead, it can be built through forward-looking, well-calibrated resource planning. As healthcare systems continue to face increasing volatility—from pandemics to climate-driven disasters—such frameworks offer both operational clarity and ethical direction. Rather than forcing a trade-off between access and cost, this approach shows how both objectives can be achieved in tandem through thoughtful, anticipatory design.

#### Model Generalizability

In this section, we demonstrate the effectiveness of our model on a wide range of model parameters. The purpose of this analysis is to demonstrate the generalizability of our model to any case study. To do so, we conduct a simulation-based analysis in which we generate 5000 problem instances by varying all model parameters and solve the resulting optimization instances using the current practice approach (CP) and using the optimization approach (Equation 14). We plot the results in Figure 7, in which we show the histogram of the simulated % reduction in the total costs of (Equation 14) over CP.

It is worth noting that (Equation 14) significantly outperforms the CP, achieving an average improvement of approximately 86%. This highlights the critical role of optimization models in enhancing bed management during pandemics.

### 3.3. Sensitivity Analysis

In this section, we evaluate the impact of key model parameters on the metrics of interest, with a particular focus on the patient rejection rate. Figure 8 illustrates how total patient rejections vary with rejection cost under different levels of bed capacity, represented by the number of in situ hospitals (left figure) and buffer capacities (right figure).

As the cost of rejecting a patient increases, which is a likely scenario during peak infection periods, the optimization model responds by significantly reducing patient rejections. However, this reduction follows a pattern of diminishing returns: it begins steadily and gradually tapers off. This occurs because the system’s capacity becomes increasingly saturated with both severely and mildly symptomatic patients. Another notable observation is that across all scenarios involving different numbers of in situ hospitals, the rejection rate initially decreases at the same rate. This trend diverges only beyond a certain cost threshold, at which point each scenario begins to show different degrees of improvement. This behavior is intuitive. When the rejection cost is relatively low, the model prioritizes minimizing logistical costs and thus does not fully utilize the additional capacity. As the cost increases, the model shifts focus toward minimizing rejections and starts to leverage the extra capacity. The most significant improvement occurs after opening the first in situ hospital. This finding reflects our study design, in which NOVA in situ hospitals are opened incrementally, starting with the facility offering the highest capacity. The National Conference Center, which provides 1000 beds, is the first to be activated (see Table 4). The patient rejection rate drops by approximately 33% after the first hospital becomes operational.

A similar trend is observed when analyzing patient rejection under varying numbers of buffer beds, which represent the availability of non-pandemic beds. This aspect of the study is important as it helps healthcare decision-makers assess the necessity of allocating non-pandemic beds during an emerging pandemic. As shown in the right figure, an increase of 2000 buffer beds results in a reduction of patient rejection cost by approximately 55%.

An important question we aim to explore in the following analysis is how changes in the cost associated with mildly symptomatic patients, which is lower than that of severe cases, affect the overall rejection rates for both mild and severe patients. This is a critical consideration because it is essential to ensure that, regardless of which cost parameter is adjusted, even one as low as the cost for mild cases, the optimization model continues to prioritize reducing patient rejection over minimizing logistical costs. As shown in Figure 9, the rejection rate for severely symptomatic patients remains at its minimal level across all values of the mild symptom cost. This suggests that the model absorbs higher logistical costs rather than compromising care for severe cases. This result is encouraging because it provides reassurance to healthcare agencies that the optimization framework consistently gives precedence to patients with severe symptoms, who contribute most significantly to overall healthcare costs.

### 3.4. Robust Planning Under Uncertain Pandemic Demand

While the baseline optimization model offers significant improvements over current practice, it assumes full knowledge of future patient demand, a condition rarely met during real-world pandemics. In practice, demand forecasts are subject to uncertainty stemming from underreporting, variant emergence, or changes in population behavior. In Section 2.3 and Section 2.4, we introduced a robust counterpart to our model, integrated with a compartmental SEIRD framework. This combination enables the prediction of future demand and, in addition, places it within a bounded uncertainty set. To that end, this section employs a simulation-based scheme to assess how demand uncertainty translates into real-world system outcomes. Specifically, we simulate 5000 stochastic demand trajectories using the SEIRD model (the values of the parameters of the SEIRD model along with the associated references are presented in Section A.1) and evaluate the realized healthcare costs under three decision strategies: current practice (CP), the average-based model, which optimizes against the average demand scenario, and the robust model. Figure 10 shows the resulting empirical cost distributions. Several insights emerge. First, both optimization-based strategies significantly outperform CP. The mean cost under CP exceeds USD 950 billion, while the average-based model lowers this to approximately USD 474 billion, and the robust model further reduces it to about USD 286 billion. This implies that robust solutions reduce total costs by around 39% and 70% over CP and average-based solutions. More importantly, the robust model exhibits the most concentrated distribution, with a sharper peak and a very thin tail, indicating a substantial reduction in cost variability. This improved consistency reflects a key strength of the robust approach: it not only reduces average cost but also limits exposure to extreme scenarios. In contrast, the CP and average-based distributions have heavier right tails, with cost realizations exceeding USD 1750 and USD 1250 billion in the worst cases (resp.). From a policy perspective, this difference in tail behavior is critical. While the average-based strategy performs well on average, it leaves the system vulnerable to rare but devastating demand spikes. The robust model, by contrast, effectively hedges against such events, delivering tighter control over financial risk and operational reliability. In probabilistic terms, it improves the upper quantiles of the cost distribution by effectively lowering the system’s value-at-risk under uncertain conditions.

#### 3.4.1. Incorporating Intervention Strategies

In this section, we present an analysis that demonstrates the effect of intervention strategies such as vaccination and lockdown policies on the healthcare system and, hence, healthcare costs. To that end, conduct a sensitivity analysis on both measures. Specifically, we introduce a vaccination rate parameter (η) into the SEIRD model (see Section A.2) and vary the transmission rate (β) to represent different levels of lockdown and social distancing. The analysis is integrated within the simulation framework in which the robust solution is considered. The mean values of the simulated total healthcare are presented in Figure 11 under different values of η and β.

As expected, both vaccination campaigns and stricter lockdown measures alleviate pressure on the healthcare system, thereby reducing total healthcare costs. For instance, a 10% increase in vaccination rate or a 10% reduction in interaction rate leads to an approximate 20% and 11% decrease in total healthcare costs (resp.).

#### 3.4.2. Policy Recommendation

This section mainly evaluates the trade-offs between the deterministic model and its robust variant, highlighting their relative performance and resilience. It helps healthcare decision-makers to answer the question of using the most relevant policy to be followed in emerging pandemics under different levels of parameter uncertainty, i.e., different levels of accuracy of predictions *r*.

Figure 12 presents a pairwise comparison between the cost of the robust strategy and the cost of the average-based strategy across increasing values of *r*. Each red diamond corresponds to the robust model’s cost under a different *r* value, with its corresponding average-case cost on the horizontal axis. The blue star represents the deterministic solution with no uncertainty (r=0), where both models coincide.

A key insight from this plot is that the cost of the robust strategy shows minimal variation with respect to *r*, while still offering substantial protection against adverse realizations. At r=0.3, for example, the robust model incurs an expected cost of roughly USD 285 million compared with about USD 413 million under the average-based model. This difference reflects the classic price of robustness: the robust model intentionally incurs slightly higher cost under average-case or deterministic conditions by conservatively allocating reserving capacity and spreading risk to guard against extreme but plausible demand surges. In other words, this is the premium paid to ensure stable system performance when uncertainty materializes. Importantly, the robustness pays off in worst-case realizations: although the average-based strategy performs well under nominal demand, it incurs a significantly higher cost when actual demand surges to the upper bound of the uncertainty set. This crossover, where planning to the mean becomes more expensive than planning conservatively, marks a critical threshold in robustness. Moreover, as *r* increases, this divergence becomes more pronounced. At r=0.8, for example, the robust model incurs a cost of just over USD 294 million, while the average-based model’s cost in the worst-case demand scenario exceeds USD 699 million. This widening cost gap underscores the growing penalty of ignoring uncertainty at higher levels of variability. The figure also reveals that all robust solutions remain consistently below the 45-degree line, confirming that for any level of uncertainty, the robust model is always less exposed to tail risk than its average-based counterpart. This positions robust optimization not merely as a theoretical safeguard but as a cost-effective strategy for real-world settings where the true demand distribution is uncertain and potentially skewed toward worst-case surges.

Together, these results suggest that the robust model offers a compelling balance between preparedness and efficiency. Although it sacrifices marginal cost in the nominal case, it avoids catastrophic outcomes when demand surges beyond forecasts, a scenario not only plausible but common in pandemic dynamics. The robust strategy effectively pays a premium (price of robustness) up front to limit downside risk, leading to more stable operations and predictable financial exposure. In settings where surge capacity is costly, healthcare access is politically sensitive, and forecasting is imperfect, such a strategy offers considerable practical value.

## 4. Conclusions and Future Directions

This paper addresses the critical challenge of hospital bed scheduling during pandemics, aiming to minimize total healthcare costs, including patient rejection and logistical expenses, under limited resource availability. Peak infection periods can overwhelm hospital capacity and increase mortality risks, which makes effective resource planning essential. We propose an optimization-based framework that incorporates three key supply strategies: first, interhospital resource sharing to ease local pressures; second, capacity expansion through in situ or field hospitals; and third, temporary reallocation of non-pandemic beds. These strategies together create a more adaptive and resilient scheduling approach. To support planning decisions, we include an SEIRD model to forecast disease progression and anticipate demand. Recognizing the uncertainty in patient inflow caused by human behavior and interactions, we extend the framework with a robust optimization formulation that maintains effectiveness under adverse or unpredictable conditions.

Given the computational complexity of our model, we develop an equivalent reformulation that reduces runtime by over 90%, greatly improving scalability. We also examine key structural properties that provide insights into optimal bed scheduling strategies. To illustrate its practical value, we conduct a case study using COVID-19 data from Northern Virginia. The results show that our method can reduce total healthcare costs by about 55% compared with current practices and highlight how strategic resource use balances patient rejection, particularly for severe cases, with logistical costs. The analysis also demonstrates the benefits of hospital resource sharing for better patient load distribution. A two-way sensitivity analysis evaluates the combined effect of rejection costs and the availability of additional resources, such as in situ hospitals and buffer beds. Finally, a simulation-based study for future pandemics shows that robust solutions consistently outperform average-based strategies by about 39%, supporting their adoption in pandemic planning.

From a policy perspective, the framework offers health authorities a practical decision-support tool to guide surge capacity planning. By linking epidemiological demand estimates with prescriptive optimization, the model provides clear triggers for when to activate field hospitals, allocate non-pandemic beds, or redistribute patients across hospitals. Adoption of such approaches could reduce reliance on ad hoc crisis responses, allowing for more transparent, evidence-based policies that balance cost efficiency with equitable access to care. At the same time, successful implementation would require coordinated governance structures and reliable data-sharing mechanisms across hospitals and agencies.

A promising direction for future research emerges from this work. While our model focuses on scheduling pandemic beds and has been extended to allow hospital-level decisions on allocating pandemic and non-pandemic beds, it does not explicitly account for emergency patients who require a different strategy to maximize admissions. Emergency cases are inherently uncertain, and modeling approaches need to consider policies for buffers or reserved beds for these patients. Therefore, it would be valuable to develop a combined model that incorporates all three patient types and optimizes allocations accordingly. Another future work could extend the model to a multi-stage stochastic setting, which would allow dynamic recourse at multiple decision points, albeit at a higher computational cost.

## Figures and Tables

**Figure 1 healthcare-13-02338-f001:**
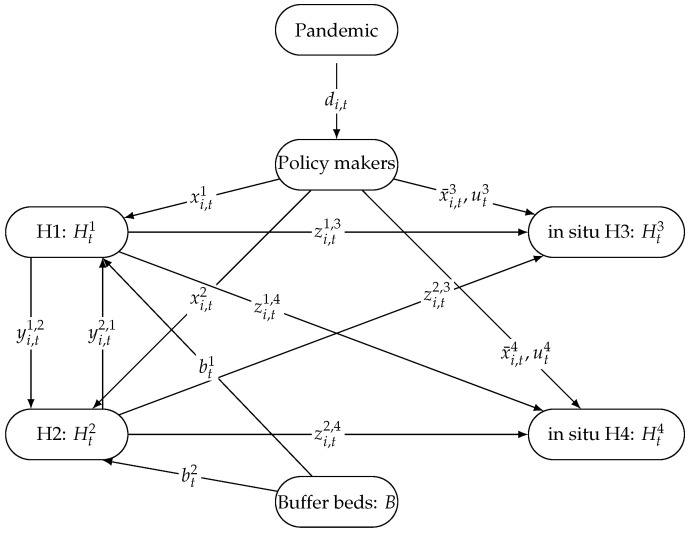
Diagram showing an illustrative example of flow of patients in a healthcare system.

**Figure 2 healthcare-13-02338-f002:**
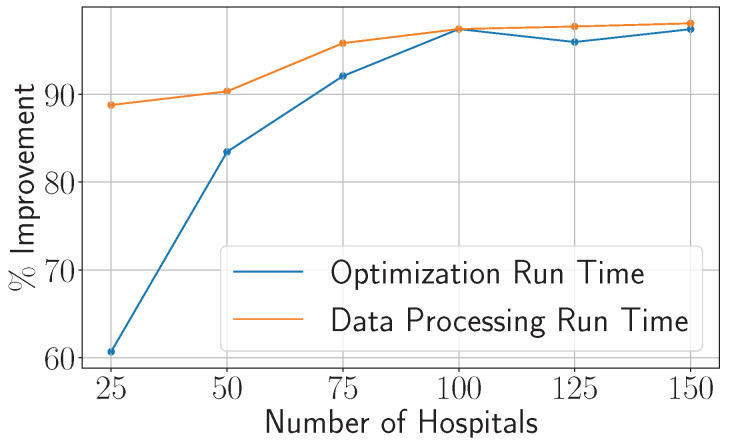
Reduction in the optimization and data processing run time as a function of problem size (number of hospitals).

**Figure 3 healthcare-13-02338-f003:**
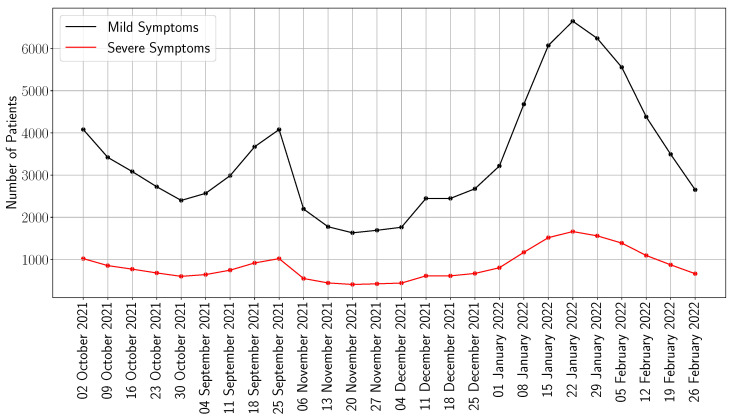
Weekly number of mild and severe cases in Northern Virginia, 2 October 2021–26 February 2022 [43].

**Figure 4 healthcare-13-02338-f004:**
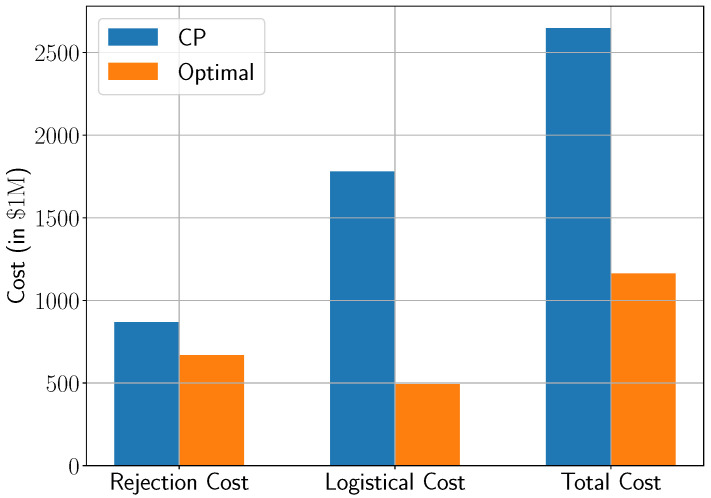
Expected healthcare costs under current practice (CP) and the proposed optimization model.

**Figure 5 healthcare-13-02338-f005:**
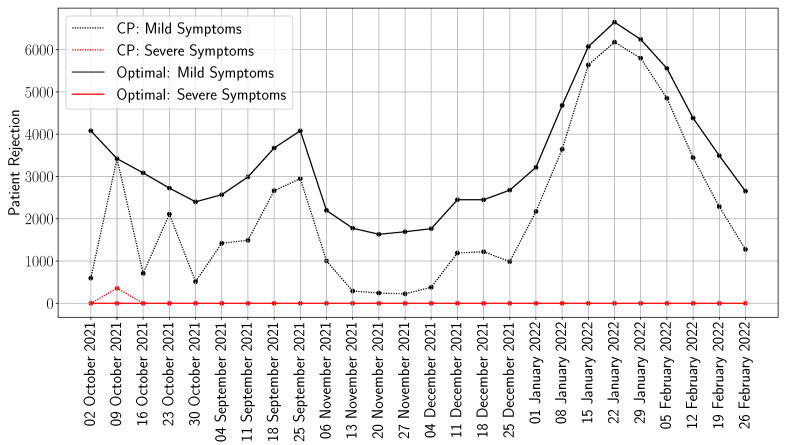
Weekly patient rejections under current practice (CP) and the optimized scheduling model, by severity of symptoms.

**Figure 6 healthcare-13-02338-f006:**
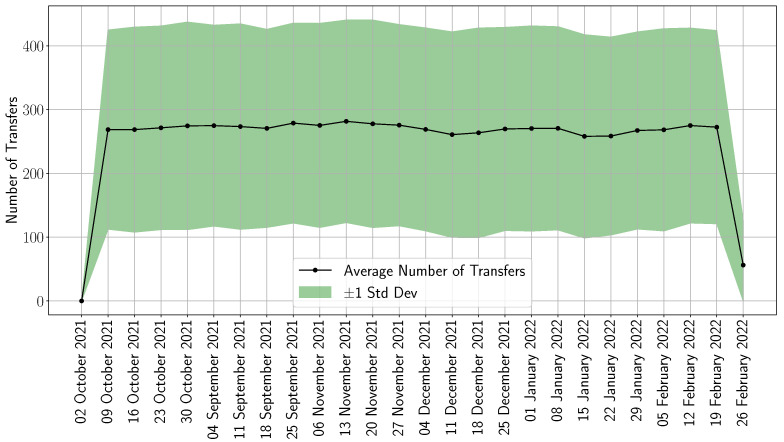
Weekly patient transfers under the optimized model. The shaded region represents ±1 standard deviation across transfer realizations.

**Figure 7 healthcare-13-02338-f007:**
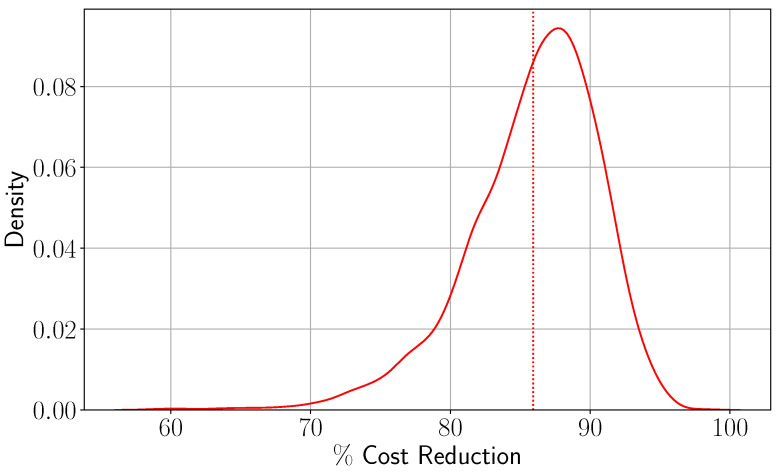
% reduction in the total costs between CP and (Equation 14) across 5000 problem instances.

**Figure 8 healthcare-13-02338-f008:**
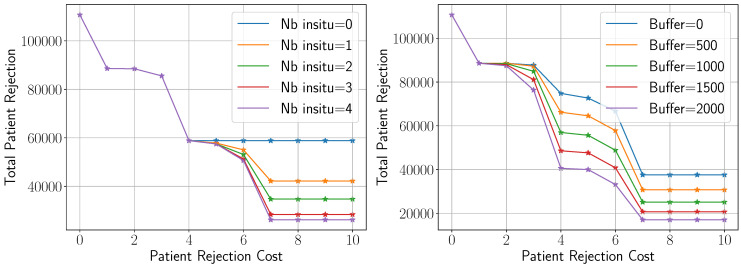
Variation of the total patient rejection as a function of rejection cost under different levels of bed capacity depicted by the number of in situ hospitals (**left**) and buffers (**right**).

**Figure 9 healthcare-13-02338-f009:**
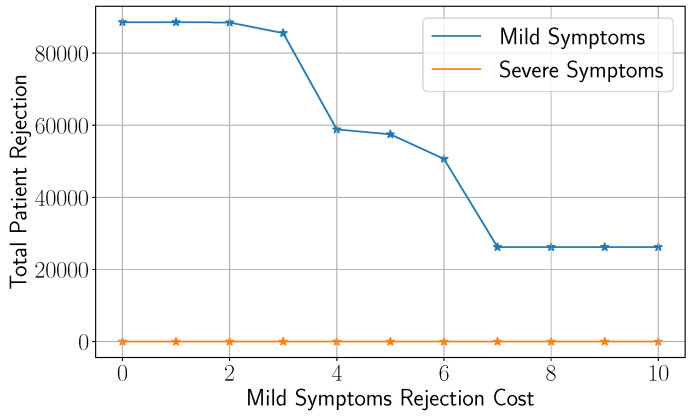
Impact of the cost of mild symptoms on the mild and severe rejection rates.

**Figure 10 healthcare-13-02338-f010:**
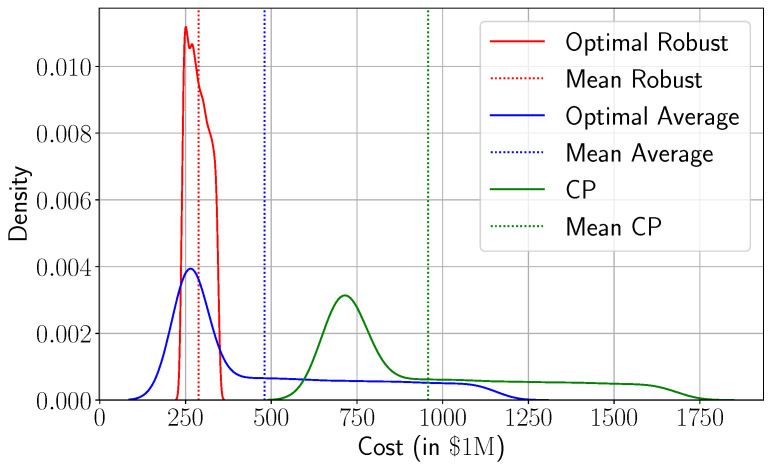
Empirical cost distributions under stochastic demand for three strategies: current practice, average-based model, and robust model. Vertical lines indicate mean costs.

**Figure 11 healthcare-13-02338-f011:**
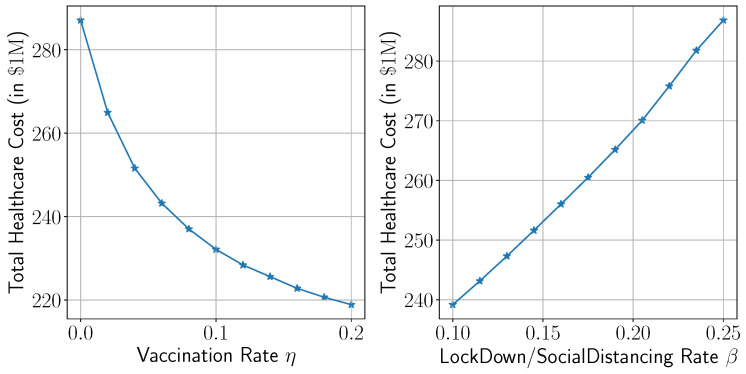
Total healthcare costs under vaccination and lockdown policies.

**Figure 12 healthcare-13-02338-f012:**
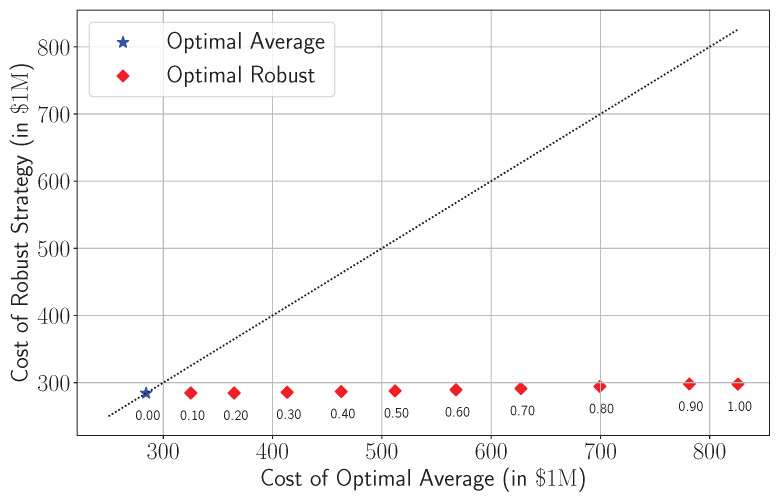
Trade-off between the cost of the average-based model and its robust counterpart as a function of the uncertainty parameter *r*. Dashed line represent the bisector line which splits the regions of the recommended policy.

**Table 1 healthcare-13-02338-t001:** Comparison of representative studies in the literature and our proposed framework.

Reference	Setting	Resources Considered	Methodology	Coordination
[2,3,4,17]	Single hospital	Bed demand (forecasting)	ML, time series	No
[7,18]	Single hospital	Beds, patient flows	Simulation/scenario analysis	No
[19]	Single hospital	ICU vs. non-pandemic beds	MDP (admission control)	No
[20]	Single hospital	COVID vs. non-COVID admissions	Dynamic programming	No
[21]	Single hospital	Operating rooms	MILP (scheduling)	No
[22,23]	Single hospital	Beds, surge strategies	Policy optimization/disposition	No
[24]	Single hospital	COVID, emergency, elective patients	Dynamic programming (multi-class)	No
[25,26]	Multi-hospital (regional)	Patient admissions	Optimization	No
[27,28,29]	Regional	Vaccines	SIR + optimization	No
[30]	Regional	Treatments	Nonlinear MIP (logistics-based)	No
[31]	Regional	Testing kits	Optimization (screening allocation)	No
[9]	Regional	Facilities, staff, transport	Integrated optimization	No
[32,33]	Regional	Ventilators	Optimization + forecasting	No
[15,34]	Regional	Beds + forecasts	Forecast + scheduling optimization	No
[11,12,13,35]	Regional	Beds, patients	Optimization/sharing frameworks	Partial
This paper	Regional (multi-hospital)	Pandemic + non-pandemic beds, in situ hospitals, buffer beds	Reformulated robust optimization integrated with SEIRD forecasts	Yes

**Table 2 healthcare-13-02338-t002:** Computational aspects of Problem (Equation 9).

Decision Variables	Type	Size
$x_{i,t}^{p}$	continuous	|Ψ|×|I|×|T|
${\bar{x}}_{i,t}^{k}$	continuous	|Ω|×|I|×|T|
$H_{t}^{p}$	continuous	|Ψ|×|T|
$H_{t}^{k}$	continuous	|Ω|×|T|
$b_{t}^{p}$	continuous	|Ψ|×|T|
$u_{t}^{k}$	binary	|Ω|×|T|
$y_{i,t}^{p,q}$	continuous	${\left|\Psi \right|}^{2}\times \left|I\right|\times \left|T\right|$
$z_{i,t}^{p,k}$	continuous	|Ψ|×|Ω|×|I|×|T|
**Constraints**	**Type**	**Size**
Equation (Equation 3)	inequality	|T|+|Ψ|×|T|
Equation (Equation 4)	inequality	|T|+|Ω|×|T|
Equation (Equation 5)	inequality	|Ω|×|T|
Equation (Equation 6)	equality	|Ψ|×|T|
Equation (7)	equality	|Ω|×|T|
Binary	-	|Ω|×|T|
Non-negativity	inequality	sum of the size of decision variables

**Table 3 healthcare-13-02338-t003:** Bed capacity of the ten major hospitals in NOVA.

Hospital Name	City	Number of Beds	Reference	Estimated Cost
Inova Fairfax Medical Campus	Falls Church	928	[47]	USD 17,234
Reston Hospital Center	Reston	243	[48]	USD 16,709
Sentara Northern Virginia Medical Center	Woodbridge	183	[49]	USD 16,884
Inova Fair Oaks Hospital	Fairfax	174	[50]	USD 16,534
Virginia Hospital Center	Arlington	343	[51]	USD 17,059
Mount Vernon Hospital	Alexandria	219	[52]	USD 16,184
Alexandria Hospital	Alexandria	318	[53]	USD 16,009
National Rehabilitation Hospital	Washington, D.C.	137	[54]	USD 15,834
StoneSprings Hospital Center	Sterling	124	[55]	USD 16,359
Westfields Hospital	Chantilly	140	[56]	USD 15,659

**Table 4 healthcare-13-02338-t004:** Bed capacity of potential in situ hospitals in NOVA.

Facility Name	City	Number of Beds	Reference	Estimated Cost
National Conference Center	Leesburg	1000	[57]	USD 32,718
Dulles Expo Center	Fairfax	500		
George Mason University	Fairfax	500		
Hilton Garden Inn	Woodbridge	200		

## Data Availability

The dataset used in this study for the NOVA COVID-19 number of infections is available [43]: https://covid.cdc.gov/COVID-data-tracker/#trends_weeklydeaths_weeklydeathratecrude_51 (accessed on 15 April 2025). All other data sources are related to our model parameters and can be found in Section 3.1, in which we provide references from published articles and/or websites.

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
