# Peer review of "An Optimization-Based Framework to Dynamically Schedule Hospital Beds in a Pandemic"

_healthcare, 2025, doi:10.3390/healthcare13182338_

Round 1

Reviewer 1 Report

Comments and Suggestions for Authors

The paper proposes an optimization framework for the dynamic management of hospital beds during a pandemic. The article is well-presented, with an appropriate writing style and high-quality tables and figures. It introduces a robust approach to the decision-making problem under study. In particular, the inclusion of a real case study strengthens the paper and increases the reader’s interest. Overall, the contribution is valuable, and I report only a few minor comments before considering the paper for possible publication in the journal:

The paper is too long, and it would be advisable to streamline some parts to improve readability. For example, the first two chapters could be merged into a single section, combining the introduction and the literature review.

The literature review would benefit from a summary table comparing the proposed work with the existing state of the art. This would help emphasize the paper’s novel contributions.

In some sections, the paper is highly technical. A small illustrative example could be introduced to clarify the type of decision problem being addressed.

I suggest adding a short subsection highlighting the managerial findings derived from the study.

Author Response

We thank the referee for the careful review and constructive comments. In the attached, we provide point-by-point responses to each of the referee’s comments.

Reviewer 2 Report

Comments and Suggestions for Authors

Dear Author,

Kindly check the attachment.

Thanks

Author Response

(The authors gave the same response as above.)
